# Achieving Ultra-Low Latency and Lossless ANN-SNN Conversion through Optimal Elimination of Unevenness Error

## Abstract

Spiking Neural Networks (SNNs) are a promising approach for neuromorphic hardware deployment due to high energy efficiency and biological plausibility. However, existing ANN–SNN conversion methods suffer notable accuracy degradation under low-latency inference, primarily caused by the *unevenness error*. To mitigate this error, prior works commonly adopt trade-off strategies at the cost of higher latency and energy consumption, such as longer time-steps, more complex spiking neuron models, or two-stage inference mechanisms. In this paper, we present a principled and efficient solution to the unevenness error. Specifically, we first develop a unified framework to quantify the unevenness error and then derive a sufficient condition for eliminating it: under an approximately constant input current, matching the ANN quantization function (floor, round, ceil) with the SNN's initial membrane potential $(0, \frac{\theta}{2}, \theta)$, where $\theta$ is the firing threshold, and setting the quantization level $L$ equals to the number of time-steps $T$, which ensures exact ANN–SNN correspondence. This finding challenges the prevailing belief that more time-steps always yield better accuracy; instead, it reveals that there exists an optimal time-step that matches the ANN's quantization characteristics, avoiding redundant inference latency from excessive time-steps. Extensive experiments on CIFAR-100, ImageNet-1K, CIFAR10-DVS, and DVS-Gesture validate our theory. For example, our method achieves a state-of-the-art 74.74% top-1 accuracy on ImageNet-1K using ResNet-34 with only 8 time-steps, demonstrating the effectiveness of our approach in low-latency SNN inference.

## 1 Introduction

Spiking neural networks (SNNs) emulate the spike-based communication of biological neurons and offer high energy efficiency on neuromorphic hardware (Maass, 1997; Merolla et al., 2014; Davies et al., 2018; DeBole et al., 2019; Pei et al., 2019). Recent advances in SNNs learning methods have enabled direct training of large-scale networks (Neftci et al., 2019). However, directly training SNNs remains challenging due to the non-differentiable nature of the spike generation. A common workaround is to use surrogate gradients (Fang et al., 2021a; Li et al., 2021b; 2022; 2024; Huang et al., 2024) to circumvent this training dilemma. As a result, the model accuracy become much inferior to counterpart ANNs and the training time is inevitably prolonged.

Alternatively, the ANN–SNN conversion paradigm provides a practical solution by transferring pretrained ANNs into SNNs, which overcomes the issues of accuracy degradation and prolonged training time. The current mainstream conversion methods map continuous activations of ANNs to the spike firing rates of SNNs (Cao et al., 2015; Han et al., 2020; Hao et al., 2023a; Bu et al., 2023; Wang et al., 2025). However, prior works require hundreds of time-steps to maintain the conversion accuracy. To address this issue, three types of error have been identified (Li et al., 2021a; Bu et al., 2023), namely: *quantization error*, *clipping error*, and *unevenness error*. Several studies have attempted to mitigate these errors. Li et al. (2021a) proposed a search-based layer-wise calibration method, but it requires tens to hundreds of long time-steps and ignores the unevenness error. Bu et al. (2022; 2023) first identified the unevenness error and introduced initial membrane-potential shift factors to drive the expected conversion error to zero; however, the performance degrades significantly at low time-steps. Hao et al. (2023a;b) proposed a two-stage method that estimates spike

offsets based on the residual membrane potential, which eliminates the unevenness error by shifting the initial membrane potential. However, the two-stage simulation introduces additional inference latency.

This paper aims to systematically address the unevenness error. We first develop a mathematical framework to quantify the unevenness error and identify its main contributing factors: the temporal distribution of input spikes, the amplitude of input currents, and the initial membrane potential of neurons. Based on this analysis, we then derive the sufficient conditions for eliminating unevenness error, establishing that the ANN quantization method (floor, round, ceil) must match the initial membrane potential $(0, \frac{\theta}{2}, \theta)$—a principle we denote as *Quantization–Voltage Matching (QVM)*, and extensive experiments verify the theory. In summary, the main contributions of this paper are as follows:

- We propose the QVM, which establishes a theoretical conversion framework for quantifying the unevenness error and derives sufficient conditions to eliminate it (**Theorem 3**).
- We challenge the prevailing concept that more time-steps always yield better conversion accuracy. Instead, there exists an optimal time-step that matches the ANN's quantization characteristics, avoiding redundant inference latency.
- We conduct experiments across CIFAR-100, ImageNet-1K, CIFAR10-DVS, and DVS-Gesture. On ImageNet-1K with ResNet-34, our method attains a top-1 accuracy of 74.74% with only 8 time-steps, enabling theoretically error-free conversion.

## 2 RELATED WORK

ANN–SNN conversion typically relies on rate coding, mapping ANN activations to spike rates. Cao et al. (2015) first studied ANN–SNN conversion by replacing ANN activations with spiking neurons. Han et al. (2020) introduced residual-membrane-potential neurons with soft resets and adaptive thresholds to reduce conversion error. Deng & Gu (2021) decomposed the conversion error into inter-layer activation mismatches and added bias compensation. Ding et al. (2021) proposed a rate-norm layer to replace ReLU activation function, and Ho & Chang (2021) introduced a trainable clip-floor activation to narrow the accuracy gap. All these works laid a solid foundation for ANN–SNN conversion but still require hundreds of inference time-steps, limiting practical deployment. Subsequent studies built on these approaches and refined them, reducing the required time-steps to dozens. Li et al. (2021a) analyzed quantization and clipping errors, calibrating activations under an assumption of uniform input currents. Bu et al. (2022) proved that setting the initial membrane potential to half the threshold can theoretically drive the expected conversion error to zero. However, both of their proofs rely on the assumption that residual membrane potentials remain bounded. Bu et al. (2023) formally defined unevenness error and proposed initial membrane potential shifting strategies, but the accuracy gaps persisted under low-latency. Hao et al. (2023a;b) further categorized unevenness error and introduced two-stage inference strategies based on residual membrane potentials, but the excessive inference stage increase both latency and overhead. Recently, Wang et al. (2025) introduced adaptive firing neuron models that search for optimal firing patterns, but this comes at the cost of increased model complexity and fails to fully eliminate the unevenness error. In summary, prior works provide valuable insights but either require excessively long time-steps or leave unevenness error unresolved. Our work addresses these limitations within a unified theoretical framework and achieves theoretically eliminate the unevenness error.

## 3 PRELIMINARY

The equations for an integrate-and-fire (IF) neuron with soft reset are as follows:

$$\boldsymbol{U}_t^l = \boldsymbol{V}_{t-1}^l + \boldsymbol{q}_t^l$$
$$\boldsymbol{s}_t^l = \mathbf{1}[\boldsymbol{U}_t^l \geq \theta^l] \tag{1}$$
$$\boldsymbol{V}_t^l = \boldsymbol{U}_t^l - \theta^l \cdot \boldsymbol{s}_t^l$$

where $\boldsymbol{U}_t^l$ is the membrane potential of the $l$-th layer neuron before firing spike, $\boldsymbol{q}_t^l = \boldsymbol{W}^l \boldsymbol{s}_t^{l-1} \theta^{l-1}$ is the input current of the $l$-th layer, $\boldsymbol{s}_t^l \in \{0, 1\}$ is the spike firing indicator function, $\boldsymbol{W}^l$ is the weight

Table 1: Summary of notations used in this paper

| Notation | Description | Notation | Description |
|---|---|---|---|
| $t$ | Time step | $\mathrm{clip}(\cdot)$ | Clipping function |
| $l$ | Layer index | $\mathcal{Q}$ | Quantization function, where $\mathcal{Q} \in \{\mathrm{floor}, \mathrm{round}, \mathrm{ceil}\}$ |
| $\boldsymbol{U}_t^l$ | Membrane potential before spiking | $\boldsymbol{a}^l$ | ANN activation value |
| $\boldsymbol{V}_t^l$ | Membrane potential after soft reset | $\boldsymbol{M}^l$ | Quantized activation value of ANN |
| $\boldsymbol{s}_t^l$ | Spike indicator function | $L$ | ANN quantization level parameter |
| $\boldsymbol{W}^l$ | Weight matrix | $\boldsymbol{\xi}^l$ | Unevenness error |
| $\theta^l$ | SNN firing threshold | $\gamma^l$ | Trainable clipping threshold of ANN |
| $\boldsymbol{q}_t^l$ | Input current at time step $t$ | $\boldsymbol{V}_0^l$ | Initial membrane potential |
| $\boldsymbol{Q}_{\mathrm{tot}}^l$ | Cumulative input current over all time steps | $\boldsymbol{z}$ | Normalized pre-synaptic input |
| $\boldsymbol{N}^l$ | Total spike count over all time steps | $L$ | ANN quantization parameter |
| $\phi^l$ | Scaled firing rate | $T$ | SNN time-steps |

matrix connecting the $(l-1)$-th layer to the $l$-th layer, and $\theta^{l-1}$ is the threshold of the $(l-1)$-th layer, which is used to scale the current formed by the input spikes.

When $\boldsymbol{U}_t^l \geq \theta^l$, the neuron fires a spike, and $\boldsymbol{V}_t^l$ is the membrane potential after soft reset following spike firing. By substituting $\boldsymbol{U}_t^l$ in equation 1, we can write the recurrence form as $\boldsymbol{V}_t^l = \boldsymbol{V}_{t-1}^l + \boldsymbol{q}_t^l - \theta^l \cdot \boldsymbol{s}_t^l$. Then, by summing over $t = 1, \ldots, T$, we can obtain the following equation:

$$\boldsymbol{V}_t^l = \boldsymbol{V}_0^l + \sum_{t=1}^{T} \boldsymbol{q}_t^l - \theta^l \sum_{t=1}^{T} \boldsymbol{s}_t^l$$
$$= \boldsymbol{V}_0^l + \boldsymbol{Q}_{tot}^l - \theta^l \cdot \boldsymbol{N}^l \tag{2}$$

where $\boldsymbol{V}_0^l$ is the initial membrane potential of the $l$-th layer neuron, $\boldsymbol{N}^l$ is the total spike count fired by the $l$-th layer neuron in $T$ time-steps, and $\boldsymbol{Q}_{tot}^l$ is the cumulative input current of the $l$-th layer neuron over $T$ time-steps.

This equation indicates that the membrane potential is conserved during the spike firing process of the IF neuron, that is: the final membrane potential is equal to the initial membrane potential plus the total input current, minus the total reset amount caused by spike firing.

The key of ANN-SNN conversion is to map the activation value $\boldsymbol{a}^l$ of the ANN using the threshold-scaled firing rate $\phi^l$ of the SNN, i.e., $\boldsymbol{a}^l \approx \phi^l$. The error arises from the deviation from the ideal relationship, where $\phi^l$ is as shown in equation 3:

$$\phi^l = \frac{\theta^l}{T} \cdot \boldsymbol{N}^l, \quad \boldsymbol{N}^l = \sum_{t=1}^{T} \boldsymbol{s}_t^l \tag{3}$$

where $\boldsymbol{N}^l \in \{0, 1, \ldots, T\}$ is the spike count over $T$ time-steps.

Ho & Chang (2021); Bu et al. (2022; 2023); Hao et al. (2023b) use a quantized activation function to train and optimize the ANN, so that quantization errors and clipping errors are absorbed into the training weights. and use the method of initial membrane potential offset to mitigate unevenness error. The clip-floor-shift equation of this method is $\boldsymbol{a}^l = \frac{\gamma^l}{L} \cdot \mathrm{clip}(\mathrm{floor}(\frac{\boldsymbol{W}^l \boldsymbol{a}^{l-1} L}{\gamma^l} + \frac{1}{2}), 0, L)$, where $L$ is the quantization level of the ANN, and $\gamma^l$ is the trainable threshold of the $l$-th layer of the ANN. Bu et al. (2023) proves that when $L$ is equal to the time-step $T$ of the SNN and $\gamma^l$ is equal to the threshold $\theta^l$ of the SNN, the conversion error is zero. However, due to the persistent unevenness error, zero-error conversion cannot be achieved. As shown in Figure 1, when $L = T$ and $\gamma^l = \theta^l$, there exists the unevenness error; when $(L \neq T, \gamma^l \neq \theta^l)$, there exist quantization error, clipping error and unevenness error.

**Motivation**: While existing ANN-SNN conversion methods have effectively mitigated quantization errors and clipping errors via trainable activation quantization strategies, they lack systematic theoretical modeling and analysis of the unevenness errors arising from the uneven temporal distribution of spikes in SNNs. This temporal unevenness error is particularly pronounced in low-time-step inference, significantly compromising conversion accuracy. This work conducts theoretical modeling and quantitative analysis of such unevenness error, and deduces the sufficient conditions for achieving zero unevenness error, thereby enabling high accuracy conversion under low-latency.

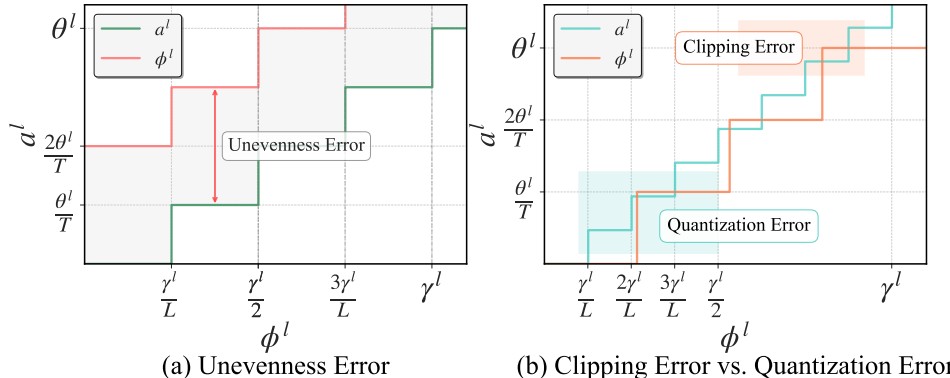

(a) Unevenness Error $\qquad$ (b) Clipping Error vs. Quantization Error

Figure 1: (a) shows the unevenness error ($L = T, \gamma^l = \theta^l$), and (b) shows the quantization error and unevenness error ($L \neq T, \gamma^l \neq \theta^l$)

## 4 UNEVENNESS ERROR IN ANN-SNN CONVERSION

In this paper, we consider a unified quantization function $\mathcal{Q} \in \{\text{floor}, \text{round}, \text{ceil}\}$, where $\text{floor}(\cdot)$ is the floor function, $\text{round}(\cdot)$ is the rounding function, and $\text{ceil}(\cdot)$ is the ceiling function. To align ANN activations with SNN spike firing rates, similar to the representations in Bu et al. (2022; 2023), we quantize the weighted activations using a trainable threshold $\gamma^l$ and a quantization level $L$, as shown in equation 4:

$$\boldsymbol{a}^l = \frac{\gamma^l}{L} \cdot \boldsymbol{M}^l, \quad \boldsymbol{M}^l = \text{clip}\left(\mathcal{Q}\left(\boldsymbol{W}^l \boldsymbol{a}^{l-1} \cdot \frac{L}{\gamma^l}\right), 0, L\right) \tag{4}$$

where $\boldsymbol{M}^l \in \{0, 1, \ldots, L\}$ is an integer tensor, indicating that the output activation of the ANN is quantized into $\boldsymbol{M}^l$ candidate values, i.e., $\boldsymbol{a}^l \in \left\{\frac{\gamma^l}{L} \cdot \boldsymbol{M}^l \big| \boldsymbol{M}^l = \{0, 1, \ldots, L\}\right\}$. The input activation of the $l$-th layer is $\boldsymbol{a}^{l-1}$. The quantization function $\mathcal{Q}$ quantizes the weighted sum of $\boldsymbol{a}^{l-1}$, and the clip function ensures that $\boldsymbol{M}^l$ is constrained within the range $[0, L]$.

Based on equation 3 and equation 4, the conversion error of the $l$-th layer in the ANN-SNN conversion is defined as: $\boldsymbol{\xi}^l = |\boldsymbol{a}^l - \boldsymbol{\phi}^l| = |\frac{\gamma^l}{L} \cdot \boldsymbol{M}^l - \frac{\theta^l}{T} \cdot \boldsymbol{N}^l|$, where $L$ is the quantization parameter of the ANN, and $T$ is the time-step of the SNN. This conversion error primarily originates from three components (Li et al., 2021a; Bu et al., 2023), detailed as follows:

**Quantization Error**: Since $\boldsymbol{\phi}^l = \frac{\theta^l}{T} \cdot \boldsymbol{N}^l$ with $\boldsymbol{N}^l \in \{0, 1, \ldots, T\}$, the value interval of $\boldsymbol{\phi}^l$ is $\frac{\theta^l}{T}$. In contrast, the activation of the ANN is $\boldsymbol{a}^l = \frac{\gamma^l}{L} \cdot \boldsymbol{M}^l$ with $\boldsymbol{M}^l \in \{0, 1, \ldots, L\}$, and its value interval is $\frac{\gamma^l}{L}$. When $L \neq T$, a mismatch between the two discrete intervals arises, giving rise to the quantization error.

**Clipping Error**: Clipping error occurs only when the upper bounds of $\boldsymbol{a}^l$ and $\boldsymbol{\phi}^l$ do not coincide. Assuming the upper bound of the quantized ANN activation $\boldsymbol{a}^l$ is $\gamma^l$. While, the SNN output $\boldsymbol{\phi}^l$ is bounded by $\theta^l$. This discrepancy leads to a range mismatch, which manifests as clipping error.

**Unevenness Error**: When $L = T$ and $\gamma^l = \theta^l$, the quantization levels of ANN activations align with the threshold-scaled firing rates of the SNN. This alignment removes other error components. However, $\boldsymbol{a}^l$ and $\boldsymbol{\phi}^l$ still do not coincide, leaving only the unevenness error, as illustrated in Figure 1. We formally define the unevenness error as:

$$\boldsymbol{\xi}^l = \left|\boldsymbol{a}^l - \boldsymbol{\phi}^l\right| = \frac{\theta^l}{T} \cdot \left|\boldsymbol{M}^l - \boldsymbol{N}^l\right| \tag{5}$$

Due to the randomness of the input spike sequence $\{s_t^{l-1}\}$, $\boldsymbol{N}^l$ may deviate from the ideal value $\boldsymbol{M}^l$, resulting in error fluctuations. Figure 3 presents the unevenness error of each layer and presents the average value of the unevenness error across all neurons in each layer. As illustrated in the figure, after adopting our proposed QVM method, the unevenness error of each layer approaches zero.

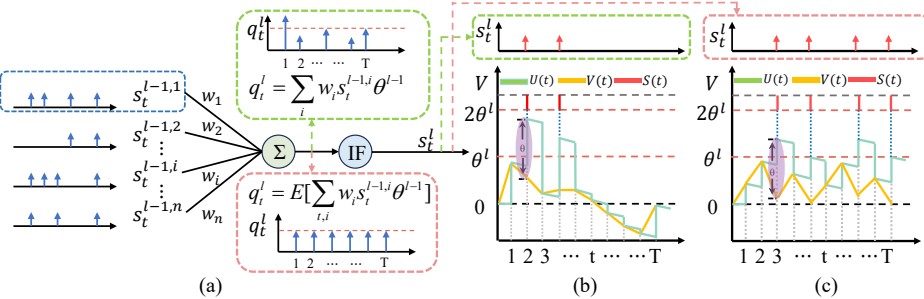

(a)  (b)  (c)

Figure 2: (a) illustrates the process of weighted summation of input spikes, where the upper part shows the uneven current and the lower part shows the uniform current; (b) depicts the process of spike accumulation and firing when the input current is uneven; (c) shows the process of spike accumulation and firing when the input current is uniform

**Factors Influencing the Spike Count** $N^l$: The membrane potential conservation equation 2 indicates that the output spike sequence $\{s_t^l\}_{t=1}^T$ of the $l$-th layer neurons depends on the initial membrane potential $V_0^l$ and time-varying input current $q_t^l = W^l s_t^{l-1} \theta^{l-1}$ and affects the residual membrane potential $V_t^l$. Specifically, $N^l$ is determined by the following three factors: (1) Temporal distribution of the input spike sequence $\{s_t^{l-1}\}_{t=1}^T$: Uneven distribution fluctuates membrane potential accumulation, deviating $N^l$ from $M^l$; (2) Input current amplitude $|q_t^l|$: Excessively large or small amplitudes cause premature or delayed firing, disrupting ideal $N^l$; (3) Initial membrane potential $V_0^l$: Inappropriate values alter threshold crossing time, leading to early or delayed spikes and mismatches between $N^l$ and $M^l$, as illustrated in Figure 2.

The conversion error can be reduced in two ways: (1) Making $N^l = M^l$ to achieve zero conversion error; (2) Increasing the number of time-steps can also make the error approach zero, i.e., $\xi^l \xrightarrow{T \to \infty} 0$, but this will lead to unacceptable inference latency.

## 5 METHOD

In this section, we first present the proof of the boundedness of the membrane potential under bounded inputs in **Theorem 1**. Building on this foundation, we propose the spike count equation in **Theorem 2**, which establishes the precise relationship between the output spike count, the initial membrane potential, and the input current. Subsequently, we derive the sufficient conditions for zero unevenness error in **Theorem 3**. A core aspect of these proofs is guaranteeing that the input current remains bounded between zero and the threshold. Finally, we present the algorithmic details and the Quantization-Voltage Matching (QVM) conversion framework.

### 5.1 BOUNDED MEMBRANE POTENTIAL UNDER BOUNDED INPUT

**Theorem 1.** *For a bounded input current $0 \le q_t^l \le \theta^l$, when the initial membrane potential $V_0^l$ satisfies $0 \le V_0^l \le \theta^l$, the membrane potential $V_t^l$ of the IF neuron after spike firing at any time $t$ always satisfies $0 \le V_t^l \le \theta^l$.*

*Proof.* The proof is by mathematical induction. The initial membrane potential $0 \le V_0^l \le \theta^l$ obviously holds. Assume that at time $t-1$, the membrane potential after spike satisfies $0 \le V_{t-1}^l \le \theta^l$. Since $0 \le q_t^l \le \theta^l$ and $0 \le V_{t-1}^l \le \theta^l$, according to equation 1: $U_t^l = V_{t-1}^l + q_t^l$, thus we have $0 \le U_t^l \le 2\theta^l$. According to whether the $U_t^l$ exceeds $\theta^l$, there are two cases: no spike firing and spike firing.

**Case 1**: If $0 \le U_t^l < \theta^l$, then $s_t^l = 0$. Thus, $0 \le V_t^l = U_t^l - \theta^l \cdot s_t^l < \theta^l$.

**Case 2**: If $\theta^l \le U_t^l \le 2\theta^l$, then $s_t^l = \mathbf{1}[U_t^l \ge \theta^l]$. Thus, $0 \le V_t^l = U_t^l - \theta^l \cdot s_t^l \le \theta^l$.

Combining **Case 1** and **Case 2**, we obtain $0 \le V_t^l \le \theta^l$. When $t = T$, the residual membrane potential satisfies $0 \le V_t^l \le \theta^l$.

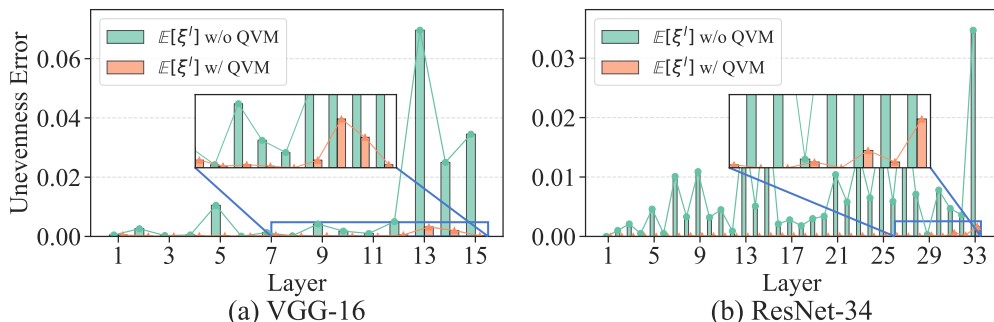

Figure 3: presents the unevenness error of VGG-16 and ResNet-34 on ImageNet-1k: green bars indicate per-layer error without the QVM method, and orange bars denote that with our QVM method. Zooming in on the near-zero data area shows the QVM method's per-layer error is close to zero.

## 5.2 SPIKE COUNT EQUATION

**Theorem 2.** *Suppose the input current is bounded by $0 \le \boldsymbol{q}_t^l \le \theta^l$ and the initial membrane potential satisfies $0 \le \boldsymbol{V}_0^l \le \theta^l$, the number of spikes within $T$ time-steps is:*

$$\boldsymbol{N}^l = \left\lfloor \frac{\boldsymbol{V}_0^l + \boldsymbol{Q}_{tot}^l}{\theta^l} \right\rfloor \tag{6}$$

*where $\boldsymbol{Q}_{tot}^l = \sum_{t=1}^{T} \boldsymbol{q}_t^l$ is the total input current and $\lfloor \cdot \rfloor$ is the floor function.*

*Proof.* From **Theorem 1** we know that $0 \le \boldsymbol{V}_t^l \le \theta^l$, and $\boldsymbol{V}_T^l$ can be discussed in two cases:

**Case 1**: $0 \le \boldsymbol{V}_t^l < \theta^l$: According to the membrane potential conservation equation 2 we have $0 \le \boldsymbol{V}_0^l + \boldsymbol{Q}_{tot}^l - \theta^l \cdot \boldsymbol{N}^l < \theta^l$, thus:

$$0 \le \boldsymbol{V}_0^l + \boldsymbol{Q}_{tot}^l - \theta^l \cdot \boldsymbol{N}^l < \theta^l \tag{7}$$

Since the number of spikes $\boldsymbol{N}^l$ must be an integer, thus $\boldsymbol{N}^l = \left\lfloor \frac{\boldsymbol{V}_0^l + \boldsymbol{Q}_{tot}^l}{\theta^l} \right\rfloor$.

**Case 2**: $\boldsymbol{V}_t^l = \theta^l$: This holds only under the extreme input condition where $\boldsymbol{V}_0^l = \theta^l$ and the input at every time-step $t$ is $\boldsymbol{q}_t^l = \theta^l$. In this case, the number of spikes can be derived from equation 2, where $\boldsymbol{Q}_{tot}^l = T \cdot \theta^l$, and the spike count is $\boldsymbol{N}^l = T$. If there exists a moment when $0 \le \boldsymbol{q}_t^l < \theta^l$, then $\boldsymbol{N}^l = \left\lfloor \frac{\boldsymbol{V}_0^l + \boldsymbol{Q}_{tot}^l}{\theta^l} \right\rfloor$ holds.

## 5.3 SUFFICIENT CONDITIONS FOR ELIMINATING UNEVENNESS ERROR

**Theorem 3.** *Under the condition that the input current is uniformly distributed across time-steps (i.e., $\boldsymbol{q}_t^l = \frac{1}{T} \cdot \boldsymbol{Q}_{tot}^l$), the ANN-SNN conversion unevenness error is eliminated ($\boldsymbol{\xi}^l = 0$) if the SNN initial membrane potential $\boldsymbol{V}_0^l$ is set according to the ANN quantization method $\mathcal{Q} \in \{\mathrm{floor}, \mathrm{round}, \mathrm{ceil}\}$ as follows:*

$$\boldsymbol{V}_0^l = \begin{cases} 0, & \mathcal{Q} = \mathrm{floor} \\ \frac{\theta^l}{2}, & \mathcal{Q} = \mathrm{round} \\ \theta^l, & \mathcal{Q} = \mathrm{ceil} \end{cases} \tag{8}$$

*Proof.* The unevenness error is defined as equation 5. Thus, proving $\boldsymbol{\xi}^l = 0$ is equivalent to proving $\boldsymbol{N}^l = \boldsymbol{M}^l$. Since the SNN time-steps $T$ equal the ANN quantization level $L$ (i.e., $T = L$) and the SNN thresholds $\theta^l$ match the ANN trainable thresholds $\gamma^l$, we use $T$ and $\theta^l$ consistently in the following derivation.

**1. Derivation of spike count under uniform input current:**

① For the SNN, the total input current over $T$ time-steps is $\boldsymbol{Q}_{\mathrm{tot}}^l = \sum_{t=1}^{T} \boldsymbol{q}_t^l$, where $\boldsymbol{q}_t^l = \boldsymbol{W}^l \boldsymbol{s}_t^{l-1} \theta^{l-1}$. However, due to the uneven distribution of the input spike sequence $\{\boldsymbol{s}_t^{l-1}\}_{t=1}^{T}$ and

the influence of weights $\boldsymbol{W}^l$, the current $\boldsymbol{q}_t^l$ can vary across time-steps, which violates the prerequisites of **Theorem 2**. Under the uniform input current condition $\boldsymbol{q}_t^l = \frac{1}{T}\boldsymbol{Q}_{\text{tot}}^l$, the prerequisite $0 \leq \boldsymbol{q}_t^l \leq \theta^l$ of **Theorem 2** is satisfied. Substituting $\boldsymbol{Q}_{\text{tot}}^l = \boldsymbol{W}^l\boldsymbol{N}^{l-1}\theta^{l-1}$ into the spike count formula equation 6, we obtain: $\boldsymbol{N}^l = \left\lfloor \frac{\boldsymbol{V}_0^l + \boldsymbol{W}^l\boldsymbol{N}^{l-1}\theta^{l-1}}{\theta^l} \right\rfloor$ ② For the ANN, the quantization level is given by equation 4. Substituting $\boldsymbol{a}^{l-1}$ with $\boldsymbol{M}^{l-1}$, we get: $\boldsymbol{M}^l = \text{clip}\left( \mathcal{Q}\left( \frac{\boldsymbol{W}^l\boldsymbol{M}^{l-1}\theta^{l-1}}{\theta^l} \right), 0, T \right)$ Let $\boldsymbol{z} = \frac{\boldsymbol{W}^l\boldsymbol{M}^{l-1}\theta^{l-1}}{\theta^l}$ denote the normalized pre-synaptic input. Since the output of the previous layer satisfies $\boldsymbol{N}^{l-1} = \boldsymbol{M}^{l-1}$, $\boldsymbol{N}^l$ and $\boldsymbol{M}^l$ can be rewritten as:

$$\boldsymbol{N}^l = \left\lfloor \frac{\boldsymbol{V}_0^l}{\theta^l} + \boldsymbol{z} \right\rfloor, \quad \boldsymbol{M}^l = \text{clip}\left( \mathcal{Q}(\boldsymbol{z}), 0, T \right) \tag{9}$$

We now verify $\boldsymbol{N}^l = \boldsymbol{M}^l$ for each quantization case. For the following derivation, $\boldsymbol{z}$ is first decomposed into an integer part $\boldsymbol{n} = \lfloor \boldsymbol{z} \rfloor$ and a fractional part $\boldsymbol{f} \in [0, 1)$, such that $\boldsymbol{z} = \boldsymbol{n} + \boldsymbol{f}$.

**2. Choice of $\boldsymbol{V}_0^l$ for quantization method $\mathcal{Q} \in \{\text{floor}, \text{round}, \text{ceil}\}$:**

**Case 1: $\mathcal{Q} = \text{floor}$ implies $\boldsymbol{V}_0^l = 0$.**

The ANN output is $\boldsymbol{M}^l = \text{clip}(\text{floor}(\boldsymbol{z}), 0, T)$, where $\text{floor}(\boldsymbol{z}) = \lfloor \boldsymbol{z} \rfloor = \boldsymbol{n}$ for $\boldsymbol{f} \in [0, 1)$. To ensure $\boldsymbol{N}^l = \boldsymbol{M}^l$, the condition: $\boldsymbol{n} \leq \frac{\boldsymbol{V}_0^l}{\theta^l} + \boldsymbol{n} + \boldsymbol{f} < \boldsymbol{n} + 1$ must hold, which simplifies to $0 \leq \frac{\boldsymbol{V}_0^l}{\theta^l} + \boldsymbol{f} < 1$. Substituting $\boldsymbol{V}_0^l = 0$ into equation 9 gives: $\boldsymbol{N}^l = \lfloor \boldsymbol{n} + \boldsymbol{f} \rfloor = \boldsymbol{n} = \lfloor \boldsymbol{z} \rfloor = \boldsymbol{M}^l$ Thus, when $\mathcal{Q} = \text{floor}$ and $\boldsymbol{V}_0^l = 0$, we have $\boldsymbol{N}^l = \boldsymbol{M}^l$ and consequently $\boldsymbol{\xi}^l = 0$.

**Case 2: $\mathcal{Q} = \text{round}$ implies $\boldsymbol{V}_0^l = \frac{\theta^l}{2}$.**

The ANN output is $\boldsymbol{M}^l = \text{clip}(\text{round}(\boldsymbol{z}), 0, T)$, where $\text{round}(\boldsymbol{z}) = \lfloor \boldsymbol{z} + 0.5 \rfloor$. This implies $\boldsymbol{M}^l = \boldsymbol{n}$ when $0 \leq \boldsymbol{f} < 0.5$ and $\boldsymbol{M}^l = \boldsymbol{n} + 1$ when $0.5 \leq \boldsymbol{f} < 1$. To ensure $\boldsymbol{N}^l = \boldsymbol{M}^l$, $\frac{\boldsymbol{V}_0^l}{\theta^l}$ must equal $0.5$, which implies $\boldsymbol{V}_0^l = \frac{\theta^l}{2}$. Substituting $\frac{\boldsymbol{V}_0^l}{\theta^l} = 0.5$ into equation 9 gives: $\boldsymbol{N}^l = \lfloor \boldsymbol{n} + \boldsymbol{f} + 0.5 \rfloor$.

- For $0 \leq \boldsymbol{f} < 0.5$, then $\boldsymbol{n} \leq \boldsymbol{n} + \boldsymbol{f} + 0.5 < \boldsymbol{n} + 1$, thus $\boldsymbol{N}^l = \boldsymbol{n} = \text{round}(\boldsymbol{z}) = \boldsymbol{M}^l$.
- For $0.5 \leq \boldsymbol{f} < 1$, then $\boldsymbol{n} + 1 \leq \boldsymbol{n} + \boldsymbol{f} + 0.5 < \boldsymbol{n} + 2$, thus $\boldsymbol{N}^l = \boldsymbol{n} + 1 = \text{round}(\boldsymbol{z}) = \boldsymbol{M}^l$.

Thus, when $\mathcal{Q} = \text{round}$ and $\boldsymbol{V}_0^l = \frac{\theta^l}{2}$, $\boldsymbol{N}^l = \boldsymbol{M}^l$ holds directly by definition.

**Case 3: $\mathcal{Q} = \text{ceil}$ implies $\boldsymbol{V}_0^l = \theta^l$.**

The ANN output is $\boldsymbol{M}^l = \text{clip}\left( \text{ceil}(\boldsymbol{z}), 0, T \right) = \lceil \boldsymbol{n} + \boldsymbol{f} \rceil$, which equals $\boldsymbol{n}$ if $\boldsymbol{f} = 0$ and $\boldsymbol{n} + 1$ if $0 < \boldsymbol{f} < 1$. Substituting $\frac{\boldsymbol{V}_0^l}{\theta^l} = 1$ into equation 9 gives: $\boldsymbol{N}^l = \lfloor 1 + \boldsymbol{n} + \boldsymbol{f} \rfloor = \boldsymbol{n} + 1 + \lfloor \boldsymbol{f} \rfloor$

- If $0 < \boldsymbol{f} < 1$, then $\lfloor \boldsymbol{f} \rfloor = 0$, so $\boldsymbol{N}^l = \boldsymbol{n} + 1$. Since $\lceil \boldsymbol{z} \rceil = \boldsymbol{n} + 1$, $\boldsymbol{N}^l = \boldsymbol{M}^l$ holds.
- If $\boldsymbol{f} = 0$ (i.e., $\boldsymbol{z}$ is an integer), $\boldsymbol{N}^l = \boldsymbol{n} + 1$ while $\boldsymbol{M}^l = \boldsymbol{n}$, creating a mismatch. However, since $\boldsymbol{z}$ is derived from weighted sums of previous layers, strictly integer values occur with negligible probability in practice.

Thus, $\boldsymbol{N}^l = \boldsymbol{M}^l$ is satisfied almost everywhere, resulting in $\boldsymbol{\xi}^l = 0$.

## 5.4 Overall Algorithm and Implementation

The detailed steps of QVM are in Algorithm 1. First, we obtain a quantized ANN where ReLU is replaced by a trainable quantization function $\mathcal{Q} \in \{\text{floor}, \text{round}, \text{ceil}\}$ with trainable clipping threshold $\gamma^l$. The QVM conversion framework then proceeds in two phases, as outlined in Algorithm 1. Phase 1: Configuration. We strictly align the SNN hyperparameters with the ANN by setting the threshold $\theta^l = \gamma^l$ and the total time-steps $T = L$, which is optimal time-steps. Crucially, the initial membrane potential $\boldsymbol{V}_0^l$ is determined by the quantization-voltage mapping defined in equation 8 (Theorem 3) to ensure error elimination. Phase 2: Inference. We employ a spike-driven simulation with uniform current distribution. The total input current derived from the previous layer's spike count is distributed equally across $T$ steps to satisfy the conditions of Theorem 3, followed by standard Integrate-and-Fire dynamics with soft-reset.

---

**Algorithm 1** Quantization-Voltage Matching (QVM) Framework

---

**Require:** Pretrained Quantized ANN $\{\boldsymbol{W}^l, \gamma^l, \mathcal{Q}\}_{l=1}^{L_{layer}}$; Quantization level $L$.
**Ensure:** SNN Time-steps $T$, SNN Parameters $\{\boldsymbol{W}^l, \theta^l, \boldsymbol{V}_0^l\}_{l=1}^{L_{layer}}$.
 1: **Initialization (QVM Setup):**
 2: Set latency $T \leftarrow L$.                                    *// optimal number of time-steps*
 3: For all layers $l \in \{1, \ldots, L_{layer}\}$:
 4:     1. Align Threshold: $\theta^l \leftarrow \gamma^l$.
 5:     2. Initialize Potential: Set $\boldsymbol{V}_0^l$ according to equation 8 based on $\mathcal{Q}$. {**Theorem 3**}
 6: **Inference (Spike-Driven Simulation):**
 7: **for** $l = 1$ **to** $L$ **do**
 8:     Calculate current according to **Theorem 3**: $\boldsymbol{q}_t^l \leftarrow \frac{\boldsymbol{Q}_{tot}^l}{T}$
 9:     **for** $t = 1$ **to** $T$ **do**
10:         $\boldsymbol{U}_t^l \leftarrow \boldsymbol{V}_{t-1}^l + \boldsymbol{q}_t^l$                    *// Membrane integration*
11:         $\boldsymbol{s}_t^l \leftarrow \mathbb{I}(\boldsymbol{U}_t^l \geq \theta^l)$                    *// Spike generation*
12:         $\boldsymbol{V}_t^l \leftarrow \boldsymbol{U}_t^l - \theta^l \cdot \boldsymbol{s}_t^l$                    *// Soft-reset*
13:     **end for**
14: **end for**

---

## 6 EXPERIMENTS

### 6.1 ABLATION STUDIES ON INITIAL MEMBRANE POTENTIAL AND QUANTIZATION METHOD

To verify **Theorem 3**, we conduct ablation experiments on CIFAR-100 using ResNet-18, with the ANN quantization level set to $L = 8$. Figure 4 shows how the SNN accuracy varies as the number of time-steps $T$ increases, when training the ANN with different quantization functions $\mathcal{Q} \in \{\text{floor}, \text{round}, \text{ceil}\}$ under three initial membrane potentials $V_0 \in \{0, \frac{\theta}{2}, \theta\}$. The gray dashed line denotes the ANN accuracy baseline, and the red vertical line marks the optimal time-step ($T = L$) that achieves the best accuracy–latency trade-off.

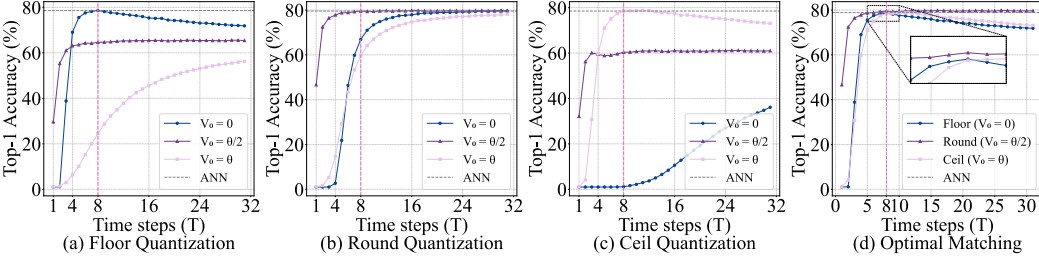

Figure 4: Different quantization methods for ANNs and initial membrane potential settings for SNNs on the variation of model accuracy with time-steps.

The details are as follows: (a) **Floor Quantization.** The highest accuracy is obtained when $V_0 = 0$, peaking at 78.62% at $T = L$, followed by a slight decline as $T$ increases. When $V_0 = \frac{\theta}{2}$ or $V_0 = \theta$, the peak accuracy is lower than that with $V_0 = 0$; although accuracy improves with larger $T$, it remains below the $V_0 = 0$ case. (b) **Round Quantization.** The best accuracy–latency trade-off occurs at $V_0 = \frac{\theta}{2}$, reaching 79.56% at $T = L$. While increasing $T$ also improves accuracy for $V_0 = \frac{\theta}{2}$ and $V_0 = \theta$, the best accuracy–latency trade-off remains at $T = L$ for $V_0 = \frac{\theta}{2}$. (c) **Ceil Quantization.** The highest accuracy is achieved when $V_0 = \theta$, peaking at 78.45% at $T = l$. For $V_0 = \frac{\theta}{2}$, accuracy varies little as $T$ increases; for $V_0 = 0$, the accuracy is lower than that with $V_0 = \theta$. (d) **Optimal matching.** According to (a)–(c), when each quantization function $\mathcal{Q}$ is matched with its corresponding initial membrane potential $V_0$ (see Eq. equation 8, i.e., the optimal matching), the accuracy curves indicate that all three quantization schemes achieve the optimal balance between accuracy and latency at $T = L$. In this case, the unevenness error is eliminated, yielding the best trade-off, and time-steps fewer or more than $L$ lead to degraded accuracy, which is consistent

with **Theorem 3**: different ANN activation quantization methods require different initial membrane potentials to maximize the recovery of SNN accuracy.

## 6.2 EXPERIMENTS ON THE IMAGENET-1K DATASET

Table 2 presents the performance of the proposed method QVM on ImageNet-1k. For VGG-16, the parameter is set to $L = 8$, and quantization function $\mathcal{Q}$ is the round. The ANN accuracy is 74.39%, and the SNN achieves the best accuracy of 74.30% when the time-step $T$ is set equal to $L$ (i.e., $T = L = 16$). At $T = 8$, QVM maintains 73.77% accuracy, slightly outperforming AdaFire (73.53%), comparable to COS (73.82% with $T + \tau = 16$), and far exceeding baselines like QCFS (19.12%) and FTBC (64.20%). Even with a reduced time-step of $T = 4$, the proposed method still reaches an accuracy of 71.20%, which is 20.07% higher than FTBC (51.13%) and 4.73% higher than SRP (66.47% with $T + \tau = 18$).

For ResNet-34, the parameter is set to $L = 8$, and ANN accuracy is 74.32%. The highest SNN accuracy of 74.74% is achieved at $T = L = 8$, surpassing all baselines including COS (74.17% with $T + \tau = 16$) and AdaFire (72.96%). When $T = 4$, QVM achieves 67.28% accuracy, outperforming FTBC by 53.58% (13.70% vs. 67.28%) and exceeding SRP (66.71% with $T + \tau = 12$) by 0.57%. At $T = 16$, QVM achieves 72.98% accuracy (higher than SRP's 68.02% and QCFS's 59.35%). When $T = 8$, it also outperforms QCFS by 39.68% (35.06% vs. 74.74%), AdaFire by 1.78% and COS (74.17% with $T + \tau = 16$) by 0.57%. These results demonstrate that the proposed method is effective on large-scale datasets, enabling high accuracy ANN-SNN conversion with low latency. The experimental results on CIFAR-100 (Appendix Table 3) also outperform previous works.

Table 2: Comparison between our method and previous works on the ImageNet-1k dataset.

| Model | Method | ANN | T | | | |
|---|---|---|---|---|---|---|
| | | | **4** | **8** | **16** | **32** |
| **VGG-16** | Calibration(Li et al., 2021a) | 75.36 | – | 25.33 | 43.99 | 62.14 |
| | SlipReLU(Jiang et al., 2023) | 71.99 | – | – | 51.54 | 67.48 |
| | QCFS(Bu et al., 2023) | 74.29 | – | 19.12 | 50.97 | 68.47 |
| | SRP* (Hao et al., 2023a) | 74.29 | 66.47 | 68.37 | 69.13 | 69.35 |
| | COS* (Hao et al., 2023b) | 74.19 | 72.94 | 73.82 | 74.09 | 74.33 |
| | FTBC(Wu et al., 2024) | 75.36 | 51.13 | 64.20 | 71.19 | 73.89 |
| | AdaFire (Wang et al., 2025) | 75.36 | – | 73.53 | 74.25 | 74.98 |
| | **QVM(L=16)** | 74.39 | 71.20 | 73.77 | **74.30** | 74.30 |
| **ResNet-34** | Calibration(Li et al., 2021a) | 75.66 | – | 0.25 | 34.91 | 61.43 |
| | SlipReLU(Jiang et al., 2023) | 75.08 | – | – | 43.76 | 66.61 |
| | QCFS(Bu et al., 2023) | 74.32 | – | 35.06 | 59.35 | 69.37 |
| | SRP* (Hao et al., 2023a) | 74.32 | 66.71 | 67.62 | 68.02 | 68.40 |
| | COS* (Hao et al., 2023b) | 74.22 | 73.81 | 74.17 | 74.14 | 73.93 |
| | FTBC(Wu et al., 2024) | 75.66 | 13.70 | 38.55 | 60.68 | 70.88 |
| | AdaFire (Wang et al., 2025) | 75.66 | – | 72.96 | 73.85 | 75.04 |
| | **QVM(L=8)** | 74.32 | 67.28 | **74.74** | 72.98 | 73.47 |

Note: Both SRP* and COS* require executing $\tau$ time-steps before inference, so the actual inference time-steps should be $T + \tau$. In SRP*, $\tau = 14$ for VGG-16 and $\tau = 8$ for ResNet-34; in COS*, $\tau = 8$ for both VGG-16 and ResNet-34.

## 6.3 ANALYSIS OF THE RELATIONSHIP BETWEEN L AND T

To further investigate how the number of SNN inference time-steps $T$ affects accuracy under different settings of the quantization level $L$, we conduct ablation experiments with VGG-16, ResNet-18, and ResNet-20 on the CIFAR-100 dataset. As shown in Figure 5, panels (a), (b), and (c) correspond to $L = 4$, $L = 6$, and $L = 8$, respectively. In accordance with **Theorem 3**, we choose the quantization function $\mathcal{Q}$ to be floor and set the initial membrane potential to $V_0 = 0$. In all panels (a)–(c), the best conversion accuracy is achieved when $T = L$. When $T \neq L$ (i.e., $T < L$ or $T > L$), the

conversion error increases. These findings challenge the common concept that larger time-steps $T$ invariably yields better conversion performance, for the following reasons.

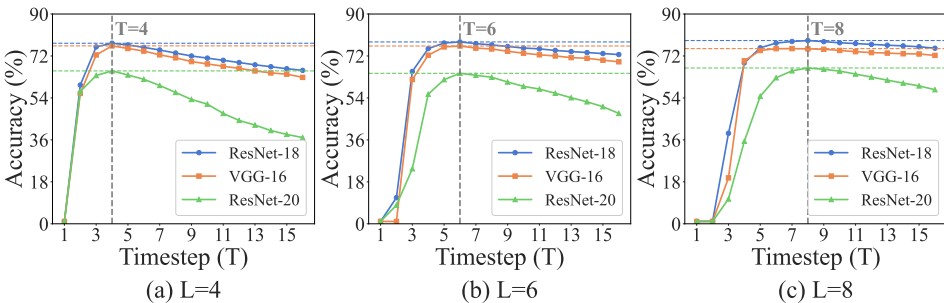

Figure 5: This figure shows the ablation experiments of ResNet-18, ResNet-20, and VGG-16 models on the CIFAR-100 dataset, illustrating the variation of SNN accuracy with time T under a fixed L

**Case 1**: When $T < L$, the expected error is given by $\boldsymbol{\xi}^l = \frac{\theta^l}{T} \cdot \left[ \frac{T}{L} \cdot \boldsymbol{M}^l - \boldsymbol{N}^l \right]$. For $T < L$, the quantization level $\boldsymbol{M}^l$ of the ANN possesses a finer resolution with L discrete levels, while the SNN's firing rate $\phi^l$ has only $T$ discrete values. Since $\frac{T}{L} < 1$, the term $\frac{T}{L} \cdot \boldsymbol{M}^l$ scales down $\boldsymbol{M}^l$, which facilitates $\boldsymbol{N}^l$ in approaching $\frac{T}{L} \cdot \boldsymbol{M}^l$. As T increases (while still remaining less than L), $\frac{T}{L} \to 1$, and $\boldsymbol{N}^l$ gets progressively closer to $\boldsymbol{M}^l$, thereby reducing the error. **Case 2**: When $T > L$, the SNN has a greater number of time-steps and a finer quantization granularity ($\frac{\theta^l}{T} < \frac{\theta^l}{L}$). In this scenario, since $\frac{T}{L} > 1$, $\frac{T}{L} \cdot \boldsymbol{M}^l$ scales up the quantized value $\boldsymbol{M}^l$ of the ANN activation, which may result in $\boldsymbol{N}^l$ failing to match accurately. The condition $T > L$ makes the SNN's quantization interval $\frac{\theta^l}{T}$ smaller than that of the ANN, and the higher resolution of the SNN might introduce additional quantization errors. Furthermore, $\frac{T}{L} \cdot \boldsymbol{M}^l$ may be a non-integer, leading to $\boldsymbol{N}^l \neq \lfloor \frac{T}{L} \cdot \boldsymbol{M}^l \rfloor$. **Case 3**: When $T = L$, the quantization intervals of the ANN and SNN are equal. As elaborated in **Theorem 3**, the unevenness error is eliminated through optimizing the expectation matching of $\boldsymbol{V}_0^l$ and $\mathcal{Q}$.

## 7 CONCLUSION

In this paper, we propose a unified theoretical framework to systematically address the *unevenness error* in ANN–SNN conversion. Unlike prior works that rely on prolonged time-steps or complex inference schemes, our approach achieves theoretically error-free conversion under low-latency settings by establishing the *Quantization–Voltage Matching (QVM)* principle. QVM aligns the ANN quantization function with the SNN's initial membrane potential and sets the number of time-steps to match the quantization level; under constant-current input activations, this eliminates the unevenness error and ensures precise ANN–SNN correspondence. Our work bridges the gap between theory and practice in ANN–SNN conversion by providing a provably optimal remedy for the unevenness error, offering both accuracy and efficiency for neuromorphic computing applications.

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

# A APPENDIX

## A.1 IMPACT OF TEMPORAL DISTRIBUTION OF SPIKES ON CONVERSION ERROR

To investigate how the temporal distribution of input spikes affects the ANN-SNN conversion error, we conducted experiments on a single-layer neural network with 1,000 neurons. The experimental procedure and settings are detailed below, and results are presented in Figure 6.

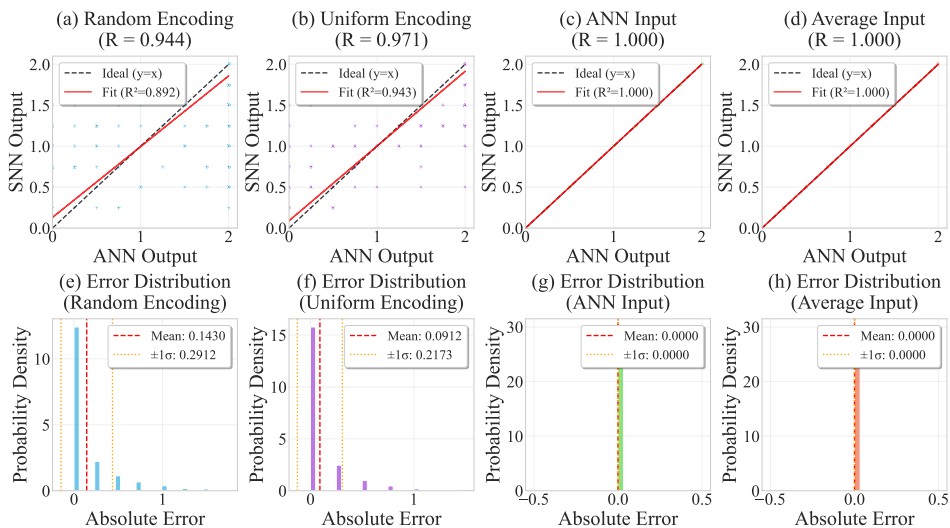

Figure 6: Impact of the temporal distribution of spikes on conversion error

**Setup 1: ANN and SNN Signal Generation**: The input activation $a^{l-1}$ of the ANN was sampled from a uniform distribution $U(0, \theta^{l-1})$. The ANN's output activation $a^l$ was computed using Eq. equation 4. Using the ANN's input $a^{l-1}$, we derived the corresponding SNN input spike count $N^{l-1} = \frac{a^{l-1} \cdot T}{\theta^{l-1}}$ (where $N^{l-1} = \sum_{t=1}^{T} s_t^{l-1}$, with $s_t^{l-1}$ denoting the spike signal at time-step t) and scaled spike firing rate $\phi^{l-1} = \frac{\theta^{l-1} \cdot N^{l-1}}{T}$.

**Setup 2: Input Spike or Currrent Scenarios**: To isolate the impact of temporal distribution, we tested four input scenarios for the SNN (under the condition $a^{l-1} = \phi^{l-1}$). For all scenarios, the membrane potential accumulation and spike firing processes followed equation 1 identically.:

- Random distribution: Spikes were randomly assigned across T time-steps.

- Uniform distribution: Spikes were equally spaced at intervals of $\frac{T}{N^{l-1}}$ within T time-steps.

- ANN input: The SNN received the ANN's output activation $a^l$ as input at each time-step.

- Constant expected current: The SNN received the constant expected current defined in **Theorem 3** at each time-step.

The experimental parameters are configured as follows: the thresholds are set to $\theta^{l-1} = 1.5$ and $\theta^l = 2.0$, which are consistent with the parameter ranges of typical SNNs; the time-step $T = 8$ is adopted to simulate low-latency inference; the weights are sampled from a uniform distribution $U(-1, 1)$; the random seed is seed $= 48$ to ensure experimental reproducibility; and each group of experiments is conducted for 1000 trials, with the results reported as the mean ± standard deviation.

The results are presented in Figure 6, with key observations as follows: Subfigures (a) and (b) illustrate the relationship between the SNN's output spike count and the ANN's output activation under random and uniform input spike sequences, respectively. Here, $R$ denotes the Pearson linear correlation coefficient, scatter plot shows that the correlation between the ANN's output activation $a^l$ and the SNN's scaled firing rate $\phi^l$ is higher under the random distribution ($R = 0.971$) than under the uniform distribution ($R = 0.944$). Subfigure (e) shows that the average absolute error under random input spikes is 0.1430, while subfigure (f) reveals that uniform spike coding achieves

superior performance with a lower average absolute error of 0.0912. Subfigures (c) and (g) display the activation alignment and error distribution when the SNN input current is derived from the ANN's output $a^l$, the correlation coefficient $R = 1.0$ and the average absolute error is 0. Similarly, subfigures (d) and (h) present the activation and error distribution under the constant expected current input, which also yields $R = 1.0$ and zero average absolute error.

Notably, under the constant expected current scenario, $\phi^l$ and $a^l$ are strictly distributed along the diagonal of the scatter plot. This indicates that the SNN's output activation exactly matches that of the ANN, and the conversion error is completely eliminated.

## A.2 IMPACT OF THE PREVIOUS LAYER'S ERROR $\delta^{l-1}$

We analyze how the error from the previous layer, denoted $\boldsymbol{\delta^{l-1}}$, propagates to the current layer. Assume there is a misalignment between the ANN activation $\boldsymbol{a}^{l-1}$ and the SNN scaled firing rate $\phi^{l-1}$, i.e., $\boldsymbol{N}^{l-1} \neq \boldsymbol{M}^{l-1}$, where $\boldsymbol{N}^{l-1}$ represents the spike count of the previous layer and $\boldsymbol{M}^{l-1}$ denotes its quantization level. The error of the previous layer is defined as $\boldsymbol{\delta}^{l-1} = \boldsymbol{N}^{l-1} - \boldsymbol{M}^{l-1}$. The total input current to the current layer is $\boldsymbol{Q}^l_{\text{tot}} = \boldsymbol{W}^l \boldsymbol{N^{l-1}} \theta^{l-1}$, which can be rewritten as:

$$\boldsymbol{Q}^l_{\text{tot}} = \boldsymbol{W}^l \theta^{l-1} \left( \boldsymbol{M}^{l-1} + \boldsymbol{\delta}^{l-1} \right) \tag{10}$$

Substituting this into equation 6 yields:

$$\boldsymbol{N}^l = \left\lfloor \frac{\boldsymbol{V}_0^l + \boldsymbol{W}^l \theta^{l-1} \left( \boldsymbol{M}^{l-1} + \boldsymbol{\delta}^{l-1} \right)}{\theta^l} \right\rfloor \tag{11}$$

Define $\boldsymbol{z}^l = \frac{\boldsymbol{W}^l \theta^{l-1} \boldsymbol{M}^{l-1}}{\theta^l}$ representing the ideal input to the quantization function and $\boldsymbol{\delta}_z^l = \frac{\boldsymbol{W}^l \theta^{l-1} \boldsymbol{\delta}^{l-1}}{\theta^l}$ denoting the propagated error term . For the specific case where the quantization function $\mathcal{Q}$ is the floor function and the initial membrane potential $\boldsymbol{V}_0^l = 0$, the quantization level of the ANN is $\boldsymbol{M}^l = \text{clip}(\text{floor}(\boldsymbol{z}^l), 0, T)$. Ignoring the boundary effect of the clipping operation (which only limits values to the range $[0, T]$ without increasing the error bound), thus:

$$\boldsymbol{M}^l = \left\lfloor \boldsymbol{z}^l \right\rfloor, \quad \boldsymbol{N}^l = \left\lfloor \boldsymbol{z}^l + \boldsymbol{\delta}_z^l \right\rfloor \tag{12}$$

The error of the $l$-th layer is $\boldsymbol{\delta}^l = \left| \boldsymbol{N}^l - \boldsymbol{M}^l \right| = \left| \left\lfloor \boldsymbol{z}^l + \boldsymbol{\delta}_z^l \right\rfloor - \left\lfloor \boldsymbol{z}^l \right\rfloor \right|$. Based on the property of the floor function, for any real numbers x and $\Delta$, $\left| \lfloor x + \Delta \rfloor - \lfloor x \rfloor \right| \leq \lfloor |\Delta| \rfloor + 1$, we derive:

$$\left| \boldsymbol{N}^l - \boldsymbol{M}^l \right| \leq \lfloor |\boldsymbol{\delta}_z^l| \rfloor + 1 \tag{13}$$

The conversion error of the current layer is defined as $\boldsymbol{\xi}^l = \frac{\theta^l}{T} \cdot \left| \boldsymbol{M}^l - \boldsymbol{N}^l \right|$, substituting $\boldsymbol{\delta}_z^l$ into the conversion error formula yields the upper bound of the current layer's error:

$$\boldsymbol{\xi}^l \leq \frac{\theta^l}{T} \cdot \left( \left\lfloor \frac{\boldsymbol{W}^l \theta^{l-1} |\boldsymbol{\delta}^{l-1}|}{\theta^l} \right\rfloor + 1 \right) \tag{14}$$

This result demonstrates that the current layer's error is determined by four factors: the previous layer's error $\boldsymbol{\delta}^{l-1}$, the weight $\boldsymbol{W}^l$, the threshold ratio $\frac{\theta^{l-1}}{\theta^l}$, and the time-step T. Notably, a zero error in the previous layer ($\boldsymbol{\delta}^{l-1} = 0$) is a necessary condition for achieving $\boldsymbol{N}^l = \boldsymbol{M}^l$ in the current layer. Consistent with this theoretical analysis, experimental results in Figure 3 confirm that when the first layer starts with $\boldsymbol{\delta}^0 = 0$, the conversion error remains zero across all subsequent layers.

## A.3 DATASETS AND EXPERIMENTAL SETUPS

CIFAR-100 (Krizhevsky et al., 2009) is a small-scale dataset with 50,000 training and 10,000 testing images, each with a spatial resolution of 32×32 pixels (3 channels) across 100 classes. Preprocessing includes standard data augmentation: random cropping, Cutout (DeVries & Taylor, 2017), and AutoAugment (Cubuk et al., 2019). ResNet-18, ResNet-20, and VGG-16 were trained on this dataset using the Stochastic Gradient Descent (SGD) optimizer (Bottou, 2012) with an initial learning rate of 0.1, momentum of 0.9, and batch size of 300. Training ran for 300 epochs with a cosine annealing scheduler (Loshchilov & Hutter, 2016) and weight decay of $5 \times 10^{-4}$.

For large-scale evaluation, we use the ILSVRC 2012 subset of ImageNet (Deng et al., 2009), containing 1,281,167 training and 50,000 testing images (resized to 224×224 pixels). Preprocessing applies the same augmentation as CIFAR-100. ResNet-34 and VGG-16 were trained here with a batch size of 128 for 300 epochs, using SGD (initial learning rate 0.1, momentum 0.9), a cosine annealing scheduler, and weight decay of $1 \times 10^{-4}$.

We also evaluate on neuromorphic datasets: DVS-CIFAR10 (Li et al., 2017), derived from CIFAR-10 via Dynamic Vision Sensor (DVS) cameras, includes 9,000 training and 1,000 testing samples (128×128 resolution) with event-driven data. DVS128 Gesture (Amir et al., 2017), capturing 11 gestures from 29 participants under varying lighting, comprises 1,342 samples (1,208 training, 134 testing) with an average duration of 6.5 ± 1.7 seconds. Both neuromorphic datasets use SpikingJelly (Fang et al., 2023) for event-to-frame integration and follow the data augmentation strategy in Hao et al. (2024). A spiking version of ResNet-18 was tested on these datasets, trained for 300 epochs with SGD (initial learning rate 0.1), cosine annealing, and weight decay of $5 \times 10^{-4}$.

## A.4 COMPARISON WITH OTHER WORKS ON CIFAR-100 DATASET

We compare our method with state-of-the-art ANN-SNN conversion methods. Table 3 shows the experimental results on CIFAR-100. For the VGG-16 model with training parameter $L = 8$, our method achieves an accuracy of 76.20% at 4 time-steps, which is 0.78% higher than SRP. It should be noted that SRP actually requires $T + \tau$ time-steps to reach 76.20% accuracy, thus needing 8 time-steps and COS achieves 76.52% needing 12 time-steps. Our work can achieve 77.00% accuracy at $T = 8$, which exceeds the accuracy of QCFS, SlipReLU, FTBC, and AdaFire at $T = 32$ time-steps. ResNet-20 is a model with very few parameters, and our method also performs excellently. With training parameter $L = 8$, our method reaches 64.57% accuracy at $T = 4$, which is much higher than FTBC's 58.08%, and achieves 68.24% accuracy at $T = 8$. For ResNet-18 with training parameter $L = 4$, our method achieves 75.61% at $T = 2$, outperforming the existing methods of SlipReLU 73.91% and QCFS 70.79%.

Table 3: Comparison between our method and previous works on CIFAR-100 dataset.

| Model | Method | ANN | T | | | | | |
| --- | --- | --- | --- | --- | --- | --- | --- | --- |
| | | | 1 | 2 | 4 | 8 | 16 | 32 |
| **VGG-16** | CalibrationLi et al. (2021a) | 77.89 | – | – | – | – | – | 73.55 |
| | QCFS(Bu et al., 2023) | 76.28 | – | 63.79 | 69.62 | 73.96 | 76.24 | 77.01 |
| | SRP* (Hao et al., 2023a) | 76.28 | 71.52 | 74.31 | 75.42 | 76.25 | 76.42 | 76.45 |
| | COS* (Hao et al., 2023b) | 76.28 | 74.24 | 76.03 | 76.26 | 76.52 | 76.77 | 76.96 |
| | SlipReLU(Jiang et al., 2023) | 68.46 | 64.21 | 66.30 | 67.97 | 69.31 | 70.09 | 70.19 |
| | FTBC(Wu et al., 2024) | 77.87 | 32.79 | 48.99 | 60.68 | 69.52 | 74.05 | 76.39 |
| | **QVM(L=8)** | 77.01 | 45.48 | 69.84 | 76.20 | 77.02 | 77.29 | 77.14 |
| **ResNet-20** | Calibration(Li et al., 2021a) | 77.16 | – | – | – | – | – | 76.32 |
| | QCFS(Bu et al., 2023) | 69.94 | – | 19.96 | 34.14 | 55.37 | 67.33 | 69.82 |
| | SRP* (Hao et al., 2023a) | 69.94 | 46.48 | 53.96 | 59.34 | 62.94 | 64.71 | 65.50 |
| | COS* (Hao et al., 2023b) | 69.97 | 59.22 | 64.21 | 65.18 | 67.17 | 69.44 | 70.29 |
| | SlipReLU(Jiang et al., 2023) | 50.79 | 48.12 | 51.35 | 53.27 | 54.17 | 53.91 | 53.11 |
| | FTBC*(Wu et al., 2024) | 81.89 | 19.96 | 38.19 | 58.08 | 71.74 | 78.80 | 81.09 |
| | **QVM(L=8)** | 68.25 | 11.39 | 43.81 | 64.57 | 68.24 | 68.41 | 68.70 |
| **ResNet-18** | SlipReLU(Jiang et al., 2023) | 74.01 | 71.51 | 73.91 | 74.89 | 75.40 | 75.41 | 75.30 |
| | QCFS(Bu et al., 2023) | 78.80 | – | 70.79 | 75.67 | 78.48 | 79.48 | 79.62 |
| | **QVM(L=8)** | 78.88 | 59.16 | 75.61 | 78.87 | 79.42 | 79.55 | 79.48 |

Both SRP* and COS* require executing $\tau$ time-steps before inference, so the actual inference time-steps should be $T + \tau$. In SRP* and COS*, $\tau = 4$. FTBC* is not a standard ResNet-20.

## A.5 COMPARISON WITH OTHER WORKS ON NEUROMORPHIC DATASET

Regarding the preprocessing of DVS data, we adopt the method from the open-source code of AdaFire [2], where the temporal dimension and feature channel dimension are merged into a sin-

gle feature channel dimension. This eliminates the explicit temporal dimension during ANN pre-training, ensuring the ANN can be trained normally. On the DVS-Gesture dataset, PLIF and CLIF achieves accuracy of 97.57% and 97.92% but requires $T = 20$; KLIF reaches 94.10% at $T = 12$. In contrast, the proposed method attains 93.75% accuracy with only $T = 4$, significantly reducing computational overhead by minimizing the time-steps. On the CIFAR10-DVS dataset, among existing methods, Dspike and DSR both operate at $T = 10$, with accuracy of 75.40% and 77.30% respectively. PLIF achieves accuracy of 74.80% but requires $T = 20$. AdaFire reduces the time-steps to $T = 8$ and increases the accuracy to 81.25%. In contrast, the proposed method with $T = 4$ reaches an accuracy of 84.50%, which is 3.25% higher than AdaFire, 7.20% higher than DSR, 9.10% higher than Dspike and 9.30% higher than PLIF.

Table 4: Comparison between our method and previous works on neuromorphic datasets.

| Dataset | Method | T | Acc.(%) |
|---|---|---|---|
| **DVS-Gesture** | PLIF(Fang et al., 2021b) | 20 | 97.57 |
| | KLIF(Jiang & Zhang, 2023) | 12 | 94.10 |
| | CLIF(Huang et al., 2024) | 20 | 97.92 |
| | **QVM(L=4)** | **4** | **93.75** |
| **CIFAR10-DVS** | PLIF(Fang et al., 2021b) | 20 | 74.80 |
| | Dspike(Li et al., 2021b) | 10 | 75.40 |
| | DSR(Meng, 2022) | 10 | 77.30 |
| | AdaFire (Wang et al., 2025) | 8 | 81.25 |
| | **QVM(L=4)** | **4** | **84.50** |

## A.6 ABLATION STUDY ON THREE QUANTIZATION METHODS UNDER DIFFERENT INITIAL MEMBRANE POTENTIALS

Table 5 presents the results of an ablation study comparing the accuracy of ResNet-18 on the CIFAR-100 dataset and $L = 8$. This study evaluates three ANN quantization methods (floor, round, ceil) when mapped to SNNs, with variables including initial membrane potential settings ($V_0 = 0$, $V_0 = \theta/2$, $V_0 = \theta$) and inference time-steps ($T = 1, 2, 4, 8, 16, 32$). Each quantization method corresponds to a unique optimal initial membrane potential: floor achieves a peak accuracy of 78.62% at $V_0 = 0$ and $T = 8$; round reaches its highest accuracy of 79.81% at $V_0 = \theta/2$ and $T = 16$; ceil attains a peak accuracy of 78.45% at $V_0 = \theta$ and $T = 8$. Notably, all methods approach or exceed their respective ANN accuracy baselines when $T = 8$. Additionally, increasing $T$ beyond 8 does not yield consistent accuracy improvements and may even cause a decline (e.g., floor quantization drops to 71.59% at $T = 32$), which validates the efficiency of the proposed $T = L$ (quantization level) setting.

Table 5: ResNet-18 accuracy comparison on CIFAR-100 dataset.

| ANN | | | SNN | | | | | |
|---|---|---|---|---|---|---|---|---|
| Method | Acc.(%) | $V_0$ | T=1 | T=2 | T=4 | T=8 | T=16 | T=32 |
| floor | 78.58 | 0 | 1.00 | 1.11 | 69.03 | **78.62** | 75.32 | 71.59 |
| | | $\theta/2$ | 29.63 | 55.28 | 63.06 | 64.60 | 65.42 | 65.25 |
| | | $\theta$ | 1.06 | 1.42 | 6.33 | 24.52 | 45.58 | 56.63 |
| round | 79.56 | 0 | 1.00 | 1.00 | 2.68 | 66.93 | 77.89 | 79.62 |
| | | $\theta/2$ | 46.62 | 72.45 | 77.81 | 79.56 | **79.81** | **79.75** |
| | | $\theta$ | 1.09 | 1.67 | 14.63 | 59.66 | 74.73 | 78.08 |
| ceil | 78.51 | 0 | 1.00 | 1.00 | 1.00 | 1.10 | 10.58 | 37.07 |
| | | $\theta/2$ | 32.07 | 56.21 | 59.43 | 60.35 | 61.07 | 61.14 |
| | | $\theta$ | 1.05 | 4.11 | 59.79 | **78.45** | 77.41 | 72.94 |

### A.7 DISCUSSION ABOUT PREVIOUS WORKS ON UNEVENNESS ERROR

Building on the concept of *unevenness error* introduced Bu et al. (2023), the authors mitigate this issue by applying an initial membrane-potential offset so that the expected conversion error is zero. However, setting the expectation conversion error to zero does not fully resolve the unevenness error. As noted by the authors, due to this error there remains a non-negligible gap between ANN and SNN accuracies even when the number of quantization levels equals the number of time steps, i.e., $L = T$. In contrast, our method eliminates the unevenness error under $L = T$, thereby enabling theoretically error-free conversion.

### A.8 THE USE OF LARGE LANGUAGE MODELS (LLMs)

Large Language Models (LLMs) are used to assist in writing, check for grammatical and spelling errors, polish language expressions, and aid in the creation of academic paper figures.

