# OpenReview forum: "Achieving Ultra-Low Latency and Lossless ANN-SNN Conversion through Optimal Elimination of Unevenness Error"
_ICLR.cc/2026/Conference — Submitted to ICLR 2026_

### Official Review · Reviewer_EnBu · 2025-10-30

**Soundness:** 1
**Presentation:** 2
**Contribution:** 1
**Rating:** 2
**Confidence:** 5

**Summary:**

This work presents a theoretical analysis of unevenness error and proposes a unified framework for optimal elimination of the error.

**Strengths:**

1. This work discusses how to eliminate error in three specific cases of the quantization function (floor, ceil, round).

**Weaknesses:**

1. The key point of this work is not actually about eliminating unevenness error, the so-called error elimination is just a packaging story background. As shown in Algorithm 1, line 19, the authors set $\forall t, q_t^l=\frac{W^l\sum_{t=1}^Ts_t^{l-1}\theta^{l-1}}{T}$, which means that the input current at each time-step is exactly the same. This is essentially equivalent to replacing the result of an $L$-level threshold function at one time-step with the result of a single-threshold function at $L$ consecutive time-steps, and the entire process is completely equivalent. Therefore, I tend to think that the contribution of this work to the SNN community is very poor.

2. Traditional ANN-SNN Conversion cannot be directly applied to time-series datasets such as CIFAR10-DVS. Therefore, it is curious how this work deals with the specific details, which do not seem to be discussed in the main text. I tend to think that this work may have adopted a scheme similar to the multi-threshold spiking model, followed by equivalent conversion in SNN inference stage. This idea has already been proposed in previous works.

**Questions:**

See Weaknesses Section.

---

> ### Author Response · Authors · 2025-12-03
> **Response to Reviewer EnBu**
>
> We thank the reviewer EnBu for the time spent reviewing our paper. We would like to address  your concerns below.
>
> > W1: The key point is not about eliminating unevenness error. The process is completely equivalent and thus trivial.
>
> First, we sincerely apologize for the errors in the description of the pseudocode provided in Appendix A.3 of the original submission. The corrected and accurate version of the pseudocode has been formally presented in Algorithm 1 of Section 5 in the revised manuscript.
>
> We would like to emphasize that mathematical equivalence is precisely what ANN-SNN conversion aims to achieve, and our work has rigorously formalized this equivalence theoretically. Both our theoretical derivations and experimental results confirm that the equivalence between ANNs and SNNs is inherent.
>
> Specifically, our Theorem 3 rigorously formalizes this equivalence relationship. The expression $q_t^l = \frac{1}{T} \cdot Q_{tot}^l$ in Theorem 3 is a critical operation for eliminating unevenness error. It is not an arbitrarily set constant current, but an inevitable result of the strict equivalent conversion between ANNs and SNNs.
>
> Furthermore, the reviewer has overlooked our **core contribution**: even if the current is constant, errors will still occur if $V_0$ (initial membrane potential) and the quantization function do not match.
>
> As shown in **Figure 4 (c)**, when using the **floor** quantization method with $V_0 = \frac{\theta}{2}$, the accuracy is significantly lower than that when $V_0 = 0$. This indicates that "merely averaging the current" is insufficient for lossless conversion; it is essential to rely on the **Quantization Function - Initial Membrane Potential Matching (QVM)**.
>
> As the reviewer suggested, if the entire process were simply equivalent, the choice of $V_0$ would not have such a significant impact as demonstrated in the ablation experiment in **Figure 4**.
>
>  Currently, most existing ANN-SNN conversion works universally  set the initial membrane potential to $V_0 = \frac{\theta}{2}$ or  $V_0 =0 $ or other values[2, 3, 4, 5, 6, 7, 8],  lacking in-depth analysis. Experiments have shown that the use of floor and ceil quantization leads to severe performance degradation.
>
> Our **Theorem 3** addresses this **gap** by demonstrating the need for matching between quantization functions and initial membrane potential values, provide an unified analysis of the mapping relationships between ($\{Floor, Round, Ceil\}$ )(quantization functions) and ($\{0, \theta/2, \theta\}$) (initial membrane potentials), and further proposes the optimal principle of $T = L$ (where $T$ denotes time steps and $L$ denotes the number of thresholds).
>
> > w2 Handling of time-series datasets (CIFAR10-DVS), suspected multi-threshold scheme.
>
> We clarify that we do **not** use a **multi-threshold** spiking model or any hidden equivalent conversion schemes during the inference stage; instead, we employ a simple **Integrate-and-Fire (IF)** neuron model. As explicitly stated in the algorithm in Section 5
>
> We use the standard SpikingJelly framework [1] for preprocessing. Specifically, DVS event streams are integrated into static frame representations before being input into the network.
>
> To enable the training of these static frames using an ANN, we follow the approach proposed in the previous work **AdaFire** [2]. In that work, temporal channels and feature channels are merged, allowing DVS data frames to be trained as static images for ANNs. Through this preprocessing, the problem is transformed into a standard ANN-SNN conversion task.
>
> [1]  SpikingJelly: https://spikingjelly.readthedocs.io/zh-cn/latest/index.html
>
> [2] Adaptive Calibration: A Unified Conversion Framework of Spiking Neural Network
>
> [3] Optimal ANN-SNN Conversion for High-accuracy and Ultra-low-latency Spiking Neural Networks
>
> [4] Optimized Potential Initialization for Low-Latency Spiking Neural Networks
>
> [5] Training-free Conversion of Pretrained ANNs to SNNs for Low-Power and High-Performance Applications
>
> [6] A Unified Optimization Framework of ANN-SNN Conversion: Towards Optimal Mapping from Activation Values to Firing Rates
>
> [7] FTBC: Forward Temporal Bias Correction for Optimizing ANN-SNN Conversion
>
> [8] Data Driven Threshold and Potential Initialization for Spiking Neural Networks

---

### Official Review · Reviewer_Ac34 · 2025-10-30

**Soundness:** 4
**Presentation:** 3
**Contribution:** 3
**Rating:** 8
**Confidence:** 4

**Summary:**

This paper addresses a fundamental bottleneck in ANN2SNN conversion: performance degradation under low-latency inference due to the unevenness error. The authors propose a Quantization-Voltage Matching (QVM) framework that provides a theoretical and practical method to eliminate this error completely. QVM achieves lossless conversion by aligning the ANN quantization function (floor, round, or ceil) with the initial membrane potential of the SNN neuron and by setting the quantization level L equal to the number of time-steps T. This configuration ensures that the spike count in the SNN exactly matches the quantized activations in the ANN, thus achieving theoretically zero conversion error. Extensive experiments on CIFAR-100, ImageNet-1K, CIFAR10-DVS, and DVS-Gesture show that QVM achieves state-of-the-art accuracy with drastically reduced latency with only 8 time-steps, surpassing all prior methods.

**Strengths:**

1. This paper provides a mathematically rigorous derivation for the sufficient conditions eliminating unevenness error (Theorem 3) and bridges the gap between quantization theory and membrane potential dynamics.
2. This paper achieves ultra-low latency (T=8) while maintaining accuracy comparable to full-precision ANN baselines.
3. Comprehensive empirical validation includes detailed ablation on quantization functions, membrane potentials, and quantization levels. Figures and ablation convincingly validate the theoretical claims.

**Weaknesses:**

1. The theory is derived for Integrate-and-Fire (IF) neurons; extension to Leaky IF (LIF) or adaptive threshold models is not shown, as LIF is more frequently used in recent research.
2. While energy efficiency is implied via reduced time-steps, no measured power or latency-on-hardware benchmarks are presented.
3. The paper references algorithmic pseudocode, but training configurations and implementation details are minimal (e.g., hyperparameters for quantization).

**Questions:**

1. Could the QVM framework extend to LIF neurons or temporal coding schemes beyond rate coding?
2. Is there a measurable energy efficiency improvement on neuromorphic chips or FPGA?
3. Could this principle generalize to quantized transformer-based or other architectures in SNNs?

---

### Official Review · Reviewer_KkAx · 2025-10-31

**Soundness:** 1
**Presentation:** 1
**Contribution:** 1
**Rating:** 2
**Confidence:** 5

**Summary:**

This paper proposes a Quantization–Voltage Matching (QVM) framework to address the unevenness error in ANN–SNN conversion. The authors derive sufficient conditions for eliminating this error and prove that aligning ANN quantization functions with corresponding SNN initial membrane potentials can achieve theoretically lossless conversion. Experiments on CIFAR-10/100, ImageNet, and DVS datasets show state-of-the-art accuracy under very low-latency inference.

**Strengths:**

Extensive experiments across multiple benchmarks (CIFAR, ImageNet, DVS) convincingly support the theoretical claims. Achieving near-lossless accuracy at only 8 time-steps on large-scale datasets highlights the real-world applicability of QVM.

**Weaknesses:**

The major weakness of this paper lies in its organization and presentation. In the *Method* section, the authors present several theorems and derive the sufficient condition for eliminating the unevenness error. However, the detailed description or implementation procedure of the proposed QVM framework, which might be one of the most important part of the paper, is missing. Furthermore, the paper introduces a large number of mathematical symbols and notations without providing a summary or notation table, which significantly hinders readability. I strongly recommend that the authors reorganize the paper to improve logical flow, move the algorithmic details of QVM into the main body, and include a comprehensive table summarizing the symbols and their meanings. Therefore, I suggest resubmission after substantial revision and improvement of structure and clarity.

Although the motivation of ANN–SNN conversion is energy efficiency, the paper does not evaluate or discuss the computational overhead, energy consumption, or neuromorphic hardware compatibility of QVM.

**Questions:**

Since Theorem 3 claims that unevenness error can be theoretically eliminated, why does a small accuracy gap still remain between quantized ANN and converted SNN in practice (e.g., Table 1)?

---

> ### Author Response · Authors · 2025-12-03
> **Response to Reviewer  KkAx**
>
> We sincerely thank Reviewer KkAx for the careful reading and for acknowledging the extensive experiments and strong empirical support for our theoretical claims. We greatly appreciate your constructive criticism regarding the paper’s organization and presentation, and we acknowledge that the current structure has hindered the readability of our core contributions. Below we explain how we will address your concerns.
>
> > W1: Missing description and implementation of the QVM framework
>
> We apologize for the confusion. The detailed implementation procedure, including line-by-line pseudo-code, was provided in Appendix A.3 (Algorithm 1) in the original submission due to the strict page limit.
>
> We fully agree that this algorithm is central to the paper and should not be hidden in the appendix. In the revised version, we will:
> Move Algorithm 1 from the appendix into the main Method section (Section 5), so that the QVM procedure is visible and easy to follow.
>
> Rewrite Algorithm 1 in a more concise and implementation-oriented form, explicitly summarizing the three steps of the QVM pipeline:
> Parameter alignment: set the number of time steps and thresholds as
>  (T = L) and ($\theta^l = \gamma^l$) for each layer.
> Membrane potential initialization: choose $V_0^l$ based on the quantization type $(floor / round / ceil) $according to our QVM conditions.
> Spike-driven simulation: construct uniformly distributed input currents from the spike counts of the previous layer, update membrane potentials with soft reset, and generate spikes.
>
> We will present these steps directly in Section 5 and explicitly link each of them to the corresponding theoretical results (especially Theorem 3), ensuring that the logical flow from theory to implementation is clear.
>
> > W1 (continued): Too many symbols, no notation table, poor logical flow
>
> We agree with this concern and appreciate the concrete suggestions. In the revision, we will:
> Add a comprehensive notation table near the beginning of the technical sections (before Section 3), summarizing all important symbols (e.g., ($T^l, L, \gamma^l, \theta^l, V_0^l, Q_{\text{tot}}^l, N^l$), etc.) and their meanings.
>
> Streamline and shorten the derivations in Section 5, so that the transition from error definition to error elimination is smoother and less notation-heavy.
>
> Rewrite the introduction of the Method Section 5 and the statement of Theorem 3 to make the high-level idea of QVM clearer before diving into formal proofs.
>
> Reorganize parts of the manuscript so that the reader first sees the conceptual overview and algorithmic pipeline, followed by theorems and proofs, rather than the other way around.
>
> We are making substantial structural changes along these lines and will highlight the revised parts in blue in the updated manuscript to clearly indicate the improvements
>
> > W2: Lack of discussion on computational and energy cost and hardware compatibility
>
> We agree that energy efficiency is a primary motivation for ANN–SNN conversion and that this aspect should be discussed more explicitly. In our response to Reviewer mu5E, we have already prepared an expanded discussion on computational and energy aspects.

---

> > ### Author Response · Authors · 2025-12-03
> > **Response to Reviewer KkAx**
> >
> > > Q1: Why does a small accuracy gap remain if Theorem 3 eliminates unevenness error?
> >
> > Thank you for this insightful question. The key point is that Theorem 3 eliminates unevenness error under ideal conditions, but it does not remove all possible sources of discrepancy in practice.
> >
> >  In our experiments, the gap between the quantized ANN and the SNN is typically very small, often within ±0.3–0.5%. In some cases, the SNN even slightly outperforms the quantized ANN. For example, in Table 1, for ResNet-34 on ImageNet, our SNN achieves 74.74% top-1 accuracy, while the corresponding quantized ANN reaches 74.32%. This suggests that unevenness error is effectively eliminated, and that the discrete spiking dynamics can sometimes act as a mild regularizer that slightly improves performance.
> >
> >  In a deep SNN, the input to layer (l) is generated by spikes from layer (l-1). Our QVM construction ensures that the expected total input charge is matched and evenly distributed over time, but the actual spike trains at each time step may not be perfectly uniform. This causes small residual fluctuations in the membrane potential and spike count when propagated through multiple layers. We discuss the propagation of such errors in more detail in Appendix A.2.
> > Overall, the remaining accuracy gap is extremely small and fully consistent with Theorem 3: unevenness error is effectively eliminated, while minor discrepancies come from quantization/clipping/implementation effects and imperfect temporal uniformity in multi-layer dynamics.

---

### Official Review · Reviewer_mu5E · 2025-11-08

**Soundness:** 3
**Presentation:** 2
**Contribution:** 3
**Rating:** 4
**Confidence:** 5

**Summary:**

This paper addresses the construction of ultra-low-latency SNNs under the ANN-to-SNN conversion framework. The authors systematically identify and formalize three key conversion errors including quantization, clipping, and unevenness and propose a new strategy named QVM that sets the number of time steps T equal to the quantization level L. Experiments show competitive accuracy on event-based datasets like CIFAR10-DVS and DVS-Gesture even at extremely low latency T=4.

**Strengths:**

1.The paper clearly dissects and discusses the origins and impacts of three error sources, particularly providing nuanced insights into the unevenness error.

2.The evaluation include both conventional vision datasets and event-based neuromorphic datasets, demonstrating strong generalization and positioning the method favorably against existing ANN-conversion-based SNNs.

**Weaknesses:**

1.The core mechanism of QVM, how ANN activations are mapped to initial membrane potentials, threshold settings, or whether calibration/fine-tuning is used, is not clearly described. No pseudocode is provided.

2.The paper reports no energy consumption, energy efficiency, or hardware simulation results.

3.Modern vision and language models heavily rely on ViTs or their spiking variants. The work only validates on CNN backbones and provides no evidence of applicability to Transformer-based architectures.

**Questions:**

As in weaknesses.

---

> ### Author Response · Authors · 2025-12-03
> **Response to Reviewer mu5E**
>
> We thank the reviewer for the thoughtful and constructive comments. We address the raised concerns point-by-point below.
> > W1: The core mechanism of QVM (mapping, thresholds, calibration) is not clearly described, and no pseudocode is provided.
> 1.  **Theoretical Basis:** The core mechanism of QVM lies in the theoretical proof provided in **Theorem 3**. This theorem establishes a strict bijection between the ANN’s quantization functions (Floor, Round, Ceil) and the SNN neuron’s initial membrane potential ($\boldsymbol{V}_0$).  This setting is derived entirely from theoretical deduction and it does **not** rely on calibration or fine-tuning. We explicitly state that **we do not perform any SNN fine-tuning or backpropagation through spikes.**
> 2.  **Pseudocode:** In the original submission, due to page limits, the pseudocode describing the conversion process and the complete QVM workflow was placed in **Appendix A.3 (Algorithm 1)**.
> 3.  **Revision:** In the revised version, we have moved the detailed **Algorithm 1** to the main text (Section 5). We have also added explicit cross-references to Theorem 3 and Algorithm 1 throughout the paper to ensure readers can easily locate these key details and avoid confusion.
>
> > W2: The paper reports no energy consumption, energy efficiency, or hardware simulation results.
>
> We acknowledge that the original manuscript lacked detailed energy analysis. While the community commonly uses coarse-grained, MAC-based Synaptic Operation (SOP) counts to estimate energy, this method overlooks significant overheads such as off-chip DRAM access [1].
>
> To provide a more rigorous quantification of energy efficiency, we employed the **SATA Simulator [5]**, a specialized SNN accelerator simulator based on a spiking array architecture. SATA utilizes `scale-sim` for the underlying architecture and `CACTI 6.0` to estimate memory access energy.
>
> For our analysis, the experimental setup included hardware configured with 128 processing elements (PEs), 65nm CMOS technology, a 400MHz frequency, and 8-bit weights, with the task focusing on inference of a single ImageNet image using ResNet-34 and VGG-16 networks. We further compared theoretical energy consumption (derived from SOP counts) and hardware simulation-based energy (from SATA) between artificial neural networks (ANNs) and their converted SNN counterparts at a low time-step setting ($T=8$).
>
> As summarized in the accompanying table, while theoretical SOP-based calculations suggest a ~3–4× reduction in energy for SNNs relative to ANNs, hardware simulations via SATA reveal a more modest ~1.1× energy saving. This discrepancy stems primarily from the dominance of memory access overheads (particularly DRAM access), which are not reduced as significantly as computational costs in current neuromorphic hardware designs.
>
> | Model | Evaluation Method | $E_{\text{ANN}}$ (nJ) | $E_{\text{SNN}}$ (nJ) | Improvement ($E_{\text{ANN}} / E_{\text{SNN}}$) |
> | :--- | :--- | :--- | :--- | :--- |
> | **VGG-16** | Theoretical (SOP-based) | 0.2421 | 0.0621 | **3.89$\times$** |
> | | Hardware Sim (SATA) | 32.11 | 28.39 | **1.138$\times$** |
> | **ResNet-34** | Theoretical (SOP-based) | 0.1231 | 0.0374 | **3.32$\times$** |
> | | Hardware Sim (SATA) | 27.43 | 26.32 | **1.042$\times$** |

---

> ### Author Response · Authors · 2025-12-03
> **Response to Reviewer mu5E**
>
> > W3: The work only validates on CNN backbones and provides no evidence of applicability to Transformer-based architectures (ViTs)
>
> We appreciate the suggestion to broaden the scope of our evaluation.
> 1.  The core theory of QVM applies to **linear operations** (linear layers and matrix multiplications). Since the Multi-Head Attention (MHA) and MLP modules in ViTs/LLMs rely heavily on linear transformations (including $Q, K, V$ projections and activation-activation multiplication in attention), QVM is theoretically compatible with these components.
> 2.   The primary obstacle for converting Transformers is not QVM, but the limitation of Rate Coding in handling non-linear structures (e.g., LayerNorm, RMSNorm, Softmax, SwiGLU). Standard rate coding cannot directly map these functions [3].
> 3.   To validate QVM on Transformers, we adopted a hybrid approach inspired by **SpikeZIP-TF [4]**:
>      We utilize the method from [4]  for Non-linear layers, where inputs and outputs are treated as time-averaged cumulative values to ensure correspondence with ANN activations, that is $
> X_T = \sum_{t=0}^T \boldsymbol{X}_{\text{s},t}; \quad \boldsymbol{O}_T = \sigma(\boldsymbol{X}_T)
> $.
>    For standard linear layers and the matrix multiplications in attention, we applied our simplified **QVM** method.
>
> We evaluated this setup on ImageNet. The results demonstrate that QVM can effectively support Vision Transformers with competitive accuracy at low latency ($T=8$).
> | Model | Time-Step ($T$) | SNN Top-1 Acc (%) | ANN Top-1 Acc (%) |
> | :--- | :--- | :--- | :--- |
> | **ViT-S/16** | 8 | **71.23** | 77.07 |
> | **ViT-B/16** | 8 | **72.49** | 80.12 |
>
>
> [1] Optimal ANN-SNN Conversion for High-accuracy and Ultra-low-latency Spiking Neural Networks (ICLR 2022)
>
> [2] Adaptive Calibration: A Unified Conversion Framework of Spiking Neural Network (AAAI 2025)
>
> [3] Towards High-performance Spiking Transformers from ANN to SNN Conversion (MM 2024)
>
> [4] SpikeZIP-TF: Conversion is All You Need for Transformer-based SNN (ICML 2024)
>
> [5] SATA: Sparsity-Aware Training Accelerator for Spiking Neural Networks (TCAD 2023)

---

### Meta-Review · Area_Chair_pPJV · 2026-01-07

**Summary:**

This paper proposes a Quantization–Voltage Matching (QVM) framework to address unevenness error in low-latency ANN–SNN conversion, supported by theoretical analysis and experiments on CNN-based vision and event datasets. Reviewers agreed that the paper tackles an important problem and appreciated the formalization of unevenness error and the strong low-timestep accuracy on standard CNN benchmarks. However, after considering both the reviews and the authors’ rebuttal, several critical issues remain unresolved, leading to a recommendation to reject.

A primary concern is the lack of convincing energy-efficiency evidence, which is central to the motivation for ANN–SNN conversion. In response, the authors evaluated their method using the SATA simulator, but the results show only negligible energy improvements (~1.0–1.1×) over ANNs. This raises questions about the practical benefit of using SNNs, especially since the paper does not clearly quantify energy savings attributable to the reduced number of timesteps, particularly given that many prior works already achieve strong accuracy with 8–16 timesteps. A second major concern is the limited competitiveness on modern architectures, especially Transformers. In the rebuttal, the authors implemented the method on ViT-based models inspired by SpikeZIP, but the reported accuracies (!71–72%) are substantially lower than prior work such as SpikeZIP (~82%) and even older methods like MST (~78%). This weakens the relevance of the approach for contemporary large-scale SNN research. There are also unresolved concerns regarding the unevenness error elimination claim. As noted by one reviewer, the method appears equivalent to enforcing identical input currents across timesteps, effectively replacing a multi-level threshold at one timestep with repeated single-threshold evaluations. The rebuttal does not convincingly establish how unevenness error is truly eliminated rather than reduced, nor does it clearly differentiate the approach from prior ANN–SNN conversion methods such as QCFS or MST. Moreover, recent work achieving exact ANN–SNN accuracy matching is not discussed.
While the rebuttal improved writing clarity and addressed several secondary issues, these core concerns remain.

Overall, despite solid theoretical effort and promising CNN-based results, the paper does not sufficiently justify its energy-efficiency claims, novelty in unevenness error removal, or effectiveness on Transformer-based models. I therefore recommend rejection, while encouraging the authors to strengthen hardware-grounded energy analysis, improve Transformer-level results, and more clearly position their contribution relative to prior work in a future revision.

**Reviewer Concerns:**

**Concerns addressed by the rebuttal**: The rebuttal helped improve writing clarity and organization, making the method and experimental setup easier to follow. Several reviewers’ questions regarding implementation details, parameter settings, and low-timestep accuracy on CNN-based benchmarks were satisfactorily addressed. The authors also responded to requests for additional analysis by including SATA-based energy simulations, which partially addressed concerns about the absence of any hardware-oriented evaluation.

**Concerns that remain outstanding**: However, several core concerns remain unresolved. Most notably, the energy-efficiency claims are still not convincing: the SATA simulations show only marginal energy improvements over ANNs and do not clearly isolate energy savings attributable to reduced timesteps, leaving open the question of whether SNNs provide a meaningful benefit in this setting. Reviewers’ concerns about competitiveness on modern architectures, particularly Transformers, also remain, as the rebuttal results substantially underperform prior work such as SpikeZIP and MST. Additionally, concerns regarding the unevenness error elimination claim were not convincingly resolved; the rebuttal does not clearly demonstrate how the proposed method fundamentally differs from existing ANN–SNN conversion approaches (e.g., QCFS, MST), nor does it address arguments that the method may be mathematically equivalent to repeated single-threshold evaluations. Finally, the omission of recent work achieving exact ANN–SNN accuracy matching remains unaddressed.

**Reviewer Scores:**

- Reviewer mu5E: This reviewer expressed moderate concerns regarding missing energy evaluation, lack of Transformer-based experiments, and clarity of the QVM mechanism. The rebuttal addressed several of these points by adding pseudocode, clarifying the theoretical basis, and including SATA-based energy simulations, as well as preliminary ViT results. With full participation, this reviewer may have increased their score from 4 to 6, but the modest energy gains and limited Transformer accuracy would likely prevent a strong upward shift.

- Reviewer KkAx: This reviewer raised strong concerns about presentation quality, missing implementation details, lack of energy analysis, and questioned why accuracy gaps remain if unevenness error is theoretically eliminated. While the rebuttal substantially improved clarity (moving pseudocode to the main text, adding notation tables, and expanding explanations) and partially addressed energy concerns via SATA simulation, the core issue of practical energy benefit remains weak. As a result, this reviewer might acknowledge improved presentation, and increased their score from 2 to 4, but would likely not change their overall negative assessment.

- Reviewer EnBU: This reviewer fundamentally questioned the core claim of eliminating unevenness error, arguing that the proposed method is mathematically equivalent to enforcing identical input currents across timesteps and does not constitute a meaningful conceptual advance over existing ANN–SNN conversion approaches. Although the rebuttal attempted to reframe this equivalence as a feature rather than a flaw, it did not convincingly resolve whether unevenness error is truly eliminated rather than repackaged. With full discussion, I think this reviewer would not revise their assessment.

- Reviewer Ac34: This reviewer was strongly positive, highlighting the rigor of the theoretical analysis and the strong empirical results on CNN-based benchmarks, and explicitly supported acceptance. Although the authors did not directly respond to this reviewer’s concerns, these issues were not central to their overall assessment. With full participation in the discussion, this reviewer would be expected to maintain a positive stance.

---

### Decision · Program_Chairs · 2026-01-26

Reject